# On the Evolution of the Optimal Design of WDS: Shifting towards the Use of a Fractal Criterion

**Juan Saldarriaga** [1,*]**, Camilo Salcedo** [2]**, María Alejandra González** [1]**, Catalina Ortiz** [3]**, Federico Wiesner** [3] **and Santiago Gómez** [3]

1 Department of Civil and Environmental Engineering, Universidad de los Andes, Bogotá 111711, Colombia
2 Department of Civil & Architectural Engineering & Mechanics, University of Arizona, Tucson, AZ 85721, USA
3 Water Supply and Sewer Systems Research Center (CIACUA), Universidad de los Andes, Bogotá 111711, Colombia
* Correspondence: jsaldarr@uniandes.edu.co; Tel.: +57-1-339-49-49 (ext. 2805)

**Abstract:** Several researchers have proposed methodologies for addressing the problem of designing optimal water distribution systems. Metaheuristic approximations are studied the most due to the vast solution space. In search of reducing computational time, the Non-dominated Sorting Genetic Algorithm II (NSGA-II) has been tested with retrofitting from the Optimal Power Use Surface (OPUS) methodology. A previous study demonstrates how OPUS significantly improves the results since it seeks to reduce energy losses in the network, in order to approximate minimum-cost designs using fewer hydraulic executions. However, more research is still needed to determine applicable hydraulic criteria that allow an enhanced comprehension of optimal designs. Therefore, this paper aims to understand the characteristics of near-optimal solutions using designs from the retrofitted OPUS/NSGA-II Pareto fronts of four distinct networks (Hanoi, Balerma, Fossolo, and Modena). Moreover, fractal characteristics of the networks' energy dissipation, flow, and diameter distribution have been analyzed for this purpose. In this way, outcomes suggest that the hydraulic gradient line box dimension in optimal designs approaches a value of two, demonstrating that objects resemble a single-plane surface. These promising results propose fractal analysis as a practical design criterion due to its hydraulic significance and low computational cost.

**Keywords:** NSGA-II; Optimal Power Use Surface (OPUS); fractal analysis; optimal design; WDS

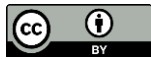

## 1. Introduction

Designing a Water Distribution System (WDS) optimally is a well-known problem and has been widely addressed by professionals worldwide. Its importance lies in supplying enough potable water to people, and doing so with limited economic resources. Shortcomings in the design process can result in deficient networks with insufficient pressure conditions to continuously supply water to all users. This problem is aggravated in developing countries due to scarce resources. For instance, a considerable percentage of WDS users receive an intermittent supply: In Africa, this amounts to a third of the users, in Asia a half, and in Latin America two-thirds [1]. Water utilities' poor strategies to adequately choose the diameters of pipes under budget limitations lead to unplanned design of water distribution systems. Consequently, pipes smaller than required are chosen to reduce investment. This situation triggers undesirable pressure conditions in the network, leaving users with insufficient water and propitiating a more rapid deterioration of infrastructure. Hence, water utilities should use efficient design techniques to better employ the available economic resources in designing functional WDSs.

In this context, a suitable strategy is necessary to minimize investments in the design process. Research has demonstrated that optimizing the energy use in the network

extends the lifespan of WDSs assets, since minimal head losses occur. Useful techniques for reducing energy losses are diameter selection during the design process and pressure control strategies for the operation. Modifying the pressure levels in the network can significantly reduce leakages and energy consumption in pumping stations [2]. Furthermore, a relevant energy use reduction can be attained by the optimal selection of diameters for pipes and pumps, seeking to employ the available energy in the most effective way [3]. The optimized energy use allows higher velocities in the system, implying that smaller pipe diameters would be required.

It is a priority that water utilities apply optimized WDS design methods and choose the most suitable commercial diameters to minimize costs while satisfying the hydraulic conditions of the network. Yet, due to the vast solution space of this problem, the optimal design of WDSs is a non-deterministic polynomial time-hard (NP-hard) problem [4]. Hence, metaheuristic approximations are necessary to achieve near-optimal solutions, such as differential evolutionary or genetic algorithms [5]. Research by Storn and Price, [6] and Savic and Walters [7] has demonstrated that these strategies have satisfactory results. However, a sizeable computational capacity is required because these strategies try to explore the entire solution space. The algorithms select random alternatives to progressively approximate better solutions based on cost or performance objective functions. Each of these alternatives must be modeled in specialized software, and usually EPANET is employed [8]. Therefore, several hydraulic executions are required to converge to a good solution, but a counterbalance must be found with the time consumed and computational resources. A strategy that researchers usually employ for this problem is to reduce the solution space using hydraulic criteria before implementing non-deterministic approaches [4,9–11].

Even though academics are constantly exploring methodologies for the optimal design of WDSs, their complexity implies difficulties, which limit their use for water utilities. Water utility resources such as time, experience, and technical knowledge play a fundamental role when choosing a design methodology. Metaheuristic approaches require an understanding of the algorithms before implementation. Thus, a limitation appears related to the calibration of the parameters, which is fundamental for the obtention of suitable results. These limitations, and the computational impediments, convert metaheuristic approximations into very expensive strategies for water utilities in terms of time and knowledge. Therefore, simple and understandable strategies are desired. Several researchers have addressed the problem, but not many straightforward methodologies can be found in the literature. Design methods based on hydraulic concepts provide advantages since they attribute physical sense to the solutions and are more understandable for utility engineers, who can make better decisions based on the specific system and the design performance. The Optimal Power Use Surface (OPUS) methodology initially proposed by Takahashi et al. [12] stands out because it is based on the minimization of head losses by the analysis of the Hydraulic Gradient Surface (HGS). Noteworthy, OPUS does not reach the global optimum design, but studies have demonstrated that it requires significantly less computational resources and is easier for engineers to apply [13].

Thus, this paper studies different methodologies for the optimal design of WDSs. Following this introduction, Section 2 presents the background of OPUS, Non-dominated Sorting Genetic Algorithm II (NSGA-II), and OPUS retrofitting NSGA-II methodologies. The box dimension concept, belonging to a branch of mathematics, is also introduced to study optimal HGSs. These methodologies, excluding the conventional application of NSGA-II, employ hydraulic concepts that should permit a better understanding and control of the optimization design process. In this way, Section 3 explains the use of each method, including the three optimization methodologies, the selection of the optimal WDS, the calculation of the fractal dimension, and the classification of WDSs. Then, the case studies are described and characterized according to their topology and functioning in Section 4. The study cases include four well-known benchmark networks: Hanoi, Fossolo, Balerma, and Modena. Next, Section 5 shows the box dimension for energy

dissipation, flow, and diameter distribution results in each case study are presented and discussed in terms of mathematical theory and hydraulic concepts. Finally, Section 6 closes the paper with the conclusions on the most outstanding outcomes from the fractal characteristics of optimal WDS designs.

## 2. Background

The methodologies that have been tested stand out for the use of criteria related to hydraulic theory for obtaining optimized designs of complex Water Distribution Systems (WDSs). The OPUS methodology appeared as an extension of the studies developed by Wu [14] and Featherstone and El-Jumaily [15]. Wu investigated the effect on the cost of different designs of a series of pipes with uniform demands. In this way, Wu identified a relation between minimum-cost designs and the target head losses of each pipe in the system, establishing a criterion based on the optimal Hydraulic Gradient Line (HGL) [14]. The optimal HGL was defined for his case study as the line with a deflection of 15% of the total available head in the middle section of a straight line connecting the source and the last node [14]. Thus, he defined optimal HGL as convex functions, which lead to minimum-cost designs. Then, Featherstone and El-Jumaily [15] extended this criterion to apply it to looped networks. The process is based on defining the target HGL for each node and the corresponding desired head-loss in each pipe, using the Euclidean distance from the reservoir as the length of pipes [15]. Instead, the OPUS methodology defines the length of pipes as the topological distance according to an optimal spanning tree defined based on a benefit-cost value, which is calculated in terms of flow and cost [13]. This process results in the optimal Hydraulic Gradient Surface (HGS), which enables the calculation of the diameters corresponding to the least-cost design.

Moreover, despite metaheuristic approximations being computationally expensive, some permit the inclusion of multiple objective functions. These are the multi-objective evolutionary algorithms (MOEAs), in which costs can be minimized while maximizing the reliability of WDSs. For this reason, the non-dominated sorting-based multi-objective evolutionary algorithm (NSGA-II) has been widely employed in WDSs' design [16,17] and it has demonstrated outstanding results compared to other MOEAs [18]. However, to make its implementation more efficient, it is necessary to implement strategies for faster convergence to near-optimal Pareto fronts (PF). These strategies can include reducing the solution space by removing non-feasible designs or initializing the algorithm with a warm start. The second strategy consists of establishing an improved initial population, as Kang and Lansey [19] did by analyzing the optimal velocities in the system, or Liu et al. [20] by determining the diameters according to the head losses in the network. Similarly, Paez et al. [5] demonstrated the improvement of the convergence rate of the conventional NSGA-II by including constant feed-back from the OPUS methodology. The process is based on computing the NSGA-II results with the optimized OPUS design every certain number of generations. The outcomes evidenced that the communication with OPUS enhances the performance of NSGA-II, especially in an early stage, making OPUS design a good energy-based warm start for approximating faster to optimal PFs.

Furthermore, the OPUS results have also been analyzed with other criteria to characterize the solutions based on the hydraulic features of least-cost designs. The surfaces generated by the topologies of networks; for example, HGS, and some other criteria demonstrate little resemblance to conventional geometric objects and seem more like fractal shapes. This has led to the study of the fractal properties of these objects. The box dimension can be observed as a measure of dispersion. In classic geometry, a point is zero dimensional, a line one dimensional, a surface is 2D and a volume is 3D. However, a set of points scattered in a volume is not any of the three dimensions. The box dimension yields a numerical value that lies between this integer dimensions and gives a sense of how similar an arbitrary object is to the conventional definitions of dimensions. This analysis has been performed considering that HGSs represent the piezometric head in each node, forming a discontinuous three-dimensional geometrical entity [13]. In this way, the

analysis has allowed for the assignment of numerical data to the complex morphology of optimal HGSs. The same procedure has been used with the total flow on a node and the sum of the diameters connected to each node. The fractal analysis therefore reveals information on the underlying structure; how disperse it is and how redundant, for example. Since the topology does not change throughout the analysis, the conclusions should be drawn around the criteria. Therefore, it has been suggested that fractal measures can be used in the design process as a fitness measure to determine the suitability of a proposed design based on preferences for redundancy or cost.

## 3. Methodology

### 3.1. OPUS

The Optimal Power Use Surface is an algorithm for optimizing WDSs based on the HGL assignment criteria developed by Wu [14]. This methodology consists of six steps, which help to understand how the available energy is used in a network [21].

The first step is to identify a Spanning Tree structure through two hydraulic principles: (1) use of a single route from the water source that supplies each node to achieve the least-cost WDS, and (2) analysis of the relationship between pipe diameter and cost per unit length [5,13,21]. The first hydraulic principle has been considered because open WDS networks are more economical than looped networks; for this reason, the main objective has been to decompose the looped system into an open tree-like structure using Prim's algorithm [22]. In the second principle, the Hazen-William equation has been implemented to determine that as the flow rate of the pipes increases, the marginal cost decreases [21]. The spanning tree starts at the water source and is expanded by adding adjacent pipe-node pairs, ensuring that the search front achieves the highest possible cost-benefit value [13]. This process has been carried out until every single node is added to the tree.

In the second step, a hydraulic head has been assigned to each node of the network according to its distance from the source and its elevation. This stage is based on the methodology proposed by Wu [14] about HGL. By assigning a minimum acceptable pressure to all the sump nodes and knowing the head of each reservoir, it is possible to implement a parabolic HGL to calculate the head of the intermediate nodes [21]. The methodology proposed by Ochoa [23] has been used to determine the optimal sag value; she found that this value depended on the cost equation, the ratio between the flow rate of a pipe and its length, and the demands at all nodes. The sag value has been calculated at each intersection using the weighted flow on each downstream course [13]. Nodes with a high elevation should be analyzed to avoid head values that do not satisfy the demanded pressure [5,13,21], the step ends when all nodes have an assigned head value.

Once the head of each node is assigned, the flow rate has been determined to calculate the diameter of each pipe in the network. This step aims to find a single flow rate that ensures mass conservation at each node and simultaneously fits the optimal power use surface calculated in the previous step [21]. The flow demand has been divided in the upstream pipes, starting from the sumps, according to one of the three criteria proposed by Saldarriaga et al. [21]:

(1) Uniform distribution: It is assumed that all pipes have the same flow, Therefore the total demand of each node is divided into the number of upstream pipes connected to it.
(2) Proportional distribution: the flow of each pipe is proportional to $H/L^2$, where H are the head losses in the pipe and L is its length.
(3) All-in-one distribution: All pipes are assigned a flow rate assuming they have the minimum diameter, and the remaining flow rate is attributed to the pipe with the highest hydraulic favorability [5,13,21].

The reliability of the network changes according to the chosen criterion. The first criterion has increased the system's reliability compared to the other two criteria [12]. The

all-in-one distribution produced the least-cost network. Therefore, the all-in-one distribution has been employed in the methodology, as it has demonstrated the best design outcomes in terms of cost [12,13].

Once the flow rate and head-losses for each pipe have been determined, the Hazen-Williams or Darcy-Weisbach equations have been implemented to compute the continuous diameters needed to fulfill the constraints. Then, these continuous diameters have been rounded to discrete diameters within the commercially available list [5,13,21]. This procedure can be conducted with different criteria; however, it was found that the best results were obtained by rounding to the nearest equivalent flow value; and this has been achieved by raising the continuous diameter values to a power of 2.6 and rounding them [21]. The last step is a post-optimization process that has two objectives: (1) to ensure that all nodes have a pressure greater than the minimum required value and (2) to seek if it is possible to reduce the construction cost of the network but still comply with all design constraints [21].

### 3.2. NSGA-II

The Non-dominated Sorting Genetic Algorithm II is a metaheuristic multi-objective optimization approach. NSGA-II has been assessed as a more efficient process for searching optimal minimum-cost and well-operating WDSs designs in a vast search space of a system. Said search space is described by the polynomial equations of energy conservation and flow variables conditioned by values contained in countable sets [18]. Yusoff et al. [24] have recapitulated the NSGA-II methodology: Initially, the algorithm starts by generating a random population from the lower to upper boundaries of some input variable; for example, the set of possible commercial diameters. Then, it performs a sorting process based on non-dominated criteria applied to the initialized population. Next, the algorithm assigns front-wise crowding distance and selects based on the rank and crowding separation. In the next stage, there is individual selection through tournament selection and crowded-comparison operation. Then, NSGA-II issues a genetic algorithm using simulated binary crossover and polynomial mutation. Finally, NSGA-II assesses the recombination of individuals of the current generation with off-spring and sets next generation's individuals by selection.

NSGA-II finds the theoretical optimal configuration of elements in a WDS that optimize the metrics defined for the obtaining of a PF. However, research has demonstrated that the sole application of the algorithm without the use of hydraulic criteria or other optimization approaches (such as sectorization techniques) produce extended computational time and complexity until arrival to the optimal configuration [10,11,25]. Moreover, NSGA-II has a slow convergence rate and produces a high number of iterations until arrival to a final WDS [13]. Furthermore, the application of an optimization procedure through NSGA-II requires an extensive parameter calibration procedure, which has not been correctly undertaken by previous WDS optimization formulations [26]. On the other hand, the OPUS methodology has found a near-optimal final WDS design by considering only the hydraulic-based principle of available energy and power use by HGL assignment at nodes and all-element calculations following hydraulic principles without a stochastic component, as opposed to NSGA-II [13]. OPUS has required far fewer iterations until a near-optimal result and has produced cost-effective WDSs designs that compare to benchmark solutions reported in the literature [14]. Therefore, there has also been interest in applying OPUS to decrease computational complexity in WDS optimization.

Paez et al. [5] have assessed an NSGA-II optimization procedure with feedback from OPUS to investigate its effect in computational time and complexity on searching for a feasible WDS optimization. Raad et al. [27] have evaluated different network reliability approaches as indicators of hydraulic performance of a WDS. The evaluation includes the Reliability Index(*RI*), the Network Reliability Index(*NRI*), and flow entropy in three benchmark networks. In addition, Creaco et al. [28] have performed a similar study that included *NRI* and *RI* assessment in WDSs. The latter study concluded that *NRI* was

applicable in both small and large networks with conclusive results in WDSs without water tanks. Furthermore, Zhan et al. [29] have evaluated reliability metrics as informers of network resilience and have concluded that *NRI* is the reliability metric best correlated with network resilience in WDSs without water tanks. Therefore, in Paez et al. [5], every WDS has gone through a bi-objective NSGA-II optimization procedure minimizing the total pipe costs and maximizing Tondini's Network Reliability Index.

The NSGA-II formulation of Paez et al. [5] has been restricted by the following constraints:

- Minimum pressure at all demand nodes.
- Maximum pressure and maximum velocity for avoidance of operational inconveniences.
- Mass and energy conservation.
- Discrete pipe diameters.

In the optimization procedure, hydraulic constraints have been met using EPANET [8] and pressure and velocity constraints have been met through cost-to-completion static penalty functions which have been activated when the design has not been able to satisfy them. Cost-to-completion functions of two kinds were used: the pressure penalty (PP) constraint penalized the objectives in the case of pressure constraints and the velocity penalty (VP) constraint penalized the objectives in the case of the velocity constraint [5]. The NSGA-II algorithm described in this paper was implemented using Matlab R2018, and connected to EPANET through the Programmer's toolkit to perform the hydraulic simulations [30].

*3.3. NSGA-II Methodology with OPUS Intermittent Feedback*

The OPUS and NSGA-II methodology is divided into three stages: (1) Calibration process, (2) preprocessing of OPUS results and (3) NSGA-II intermittent retrofitting through OPUS.

3.3.1. Calibration Process

The calibration process consists of determining the set of parameters that produce the best approximation to the benchmark PF found in the literature for each network. The ranges of values for each parameter have been calibrated as those producing the benchmark PFs.

The parameters calibrated for NSGA-II were: population size, number of generations, the crossover distribution index, the crossover probability, the mutation distribution index, the mutation probability, the parameter that establishes the magnitude of the pressure, the velocity penalization ($P_{ratio}$) and the feedback frequency.

For each parameter, multiple values within the range have been tested and; considering the dispersion and distribution of the PFs, the one with the highest convergence to the optimal PF has been chosen. The range of values used for each parameter is mentioned in the work of Paez et al. [5]. After testing multiples values for each of the parameters, the values that were used by Paez et al. [5] to run NSGA-II and retrofitted approach are shown in Table 1.

**Table 1.** Values of each parameter for the implementation of the NSGA-II and retrofitted approach. Source: Paez et al. [5].

| Network | Individuals | Generations | Mutation Distribution Index | Crossover Distribution Index | Retrofitted Frequency |
|---------|-------------|-------------|-----------------------------|------------------------------|-----------------------|
| Hanoi | 500 | 500 | 20 | 3 | 5 |
| Fossolo | 500 | 500 | 100 | 10 | 20 |
| Balerma | 2000 | 4500 | 100 | 2 | 10 |
| Modena | 2000 | 4000 | 20 | 7 | 50 |

3.3.2. Preprocessing of OPUS Results

The preprocessing step has taken place after running the first step in OPUS up to the assignment of continuous diameter to each pipe. That diameter configuration helps to determine the unit headloss ($S_f$) through a hydraulic simulation.

Using the values of unit headloss and diameter of each pipe in the system, an envelope curve for the $S_f$ vs $D$ relationship has been determined, as shown in Equation (1).

$$s_{fmax} = f_1(D) \tag{1}$$

Then, a friction function that relates flow rate capacity (Q) to the pipe diameter and the unit head-losses has been used, as shown in Equation (2).

$$Q = f_2(D, S_f) \tag{2}$$

Finally, a function is calculated to determine the diameter given a flow rate, as shown in Equation (3).

$$Q = f_2\big(D, f_1(D)\big) = f_3(D) \tag{3}$$

3.3.3. NSGA-II Intermittent Retrofitting through OPUS

The retrofitting process, i.e., the inclusion of the OPUS methodology in the NSGA-II optimization procedure, has been established from the algorithm's initialization using each diameter configuration as individuals [5]. An individual's performance has been calculated through the cost function and the *NRI* function [5]. The constraints of minimum pressure at all demand nodes have been revised, as well as the velocity constraint in the required cases. Each individual's diameter configuration has been simulated for constraint revisions in EPANET [27].

For a whole generation of NSGA-II individuals, an OPUS-recommended WDS configuration for each individual has been calculated. NSGA-II crossover and mutation steps have not been applied to update all individuals in a generation. For instance, the following generation for applying the NSGA-II methodology has been updated as a combination of NSGA-II recommended diameters and diameters obtained through the OPUS methodology using the resulting flow rates of the preliminary optimization process network as a basis combined with Equation (4) [5]. Through inspection, it has been found that the combination producing a PF closer to the best PF in the literature is obtained through a simple average between the diameters obtained through NSGA-II and the diameters obtained through the OPUS feedback process, as observed in Equation (4):

$$\text{offspring individual} = \frac{1}{2} \begin{vmatrix} d_1 + f_3^{-1}(Q_1) \\ d_2 + f_3^{-1}(Q_2) \\ \vdots \\ d_{NP} + f_3^{-1}(Q_{NP}) \end{vmatrix} \tag{4}$$

In Equation (4), $d_i$ represents a pipe diameter of an individual obtained through NSGA-II optimization and $f_3^{-1}(Q_i)$ represents a pipe diameter of an individual obtained through OPUS with flow rates extracted from the EPANET hydraulic simulation with

former individuals having diameters $d_i$ [5]. Finally, Paez et al. [5] have set a feedback frequency of $m \in$ (5, 50) for balancing a trade-off between PF quality and the algorithm's convergence rate.

### 3.4. Optimal WDSs Selection

Through the application of the NSGA-II optimization with feedback from OPUS, several PFs have been obtained. All PFs have discriminated WDS configuration solutions, only showing those that guarantee the highest contrast between cost *(C)* and Network Resilience Index. Through inspection, it has been found that the higher the number of generations, the closer the obtained PFs are to the benchmark PFS for the WDSs found in the literature. A closer approach to the benchmark PFs of the considered case studies symbolizes a greater efficiency of the retrofitting algorithm. The higher efficiency is explained by the fact that a closer resemblance to the PFs obtained through a traditional NSGA-II optimization guarantees that the developed methodology is giving feasible WDSs optimizations closer to the theoretical optimal, which correspond to the outcomes of the NSGA-II algorithm.

Hence, even if the algorithm considered different values for the number of generations for each WDS of study, only optimization outcomes for the highest number of generations have been considered in the further steps of the methodology. At this point, the solutions consist of retrofitted OPUS/NSGA-II PFs. One of the inconveniences of bi-objective optimization approaches is the lack of an impartial selection criterion of an optimized WDS between all arrangements of network elements that give the highest contrast between evaluation functions. Therefore, at this point, the engineer should choose a WDS arrangement subjectively. The further steps of the methodology aim to override subjectivity in the selection of the optimized WDS layout through the study of its fractal characteristics.

An important criterion will be the fractal dimension (*D*), as it can serve as a performance indicator in terms of hydraulic behavior of contrasting networks in terms of cost (*C*) and reliability (*NRI*) [31]. The idea is to compare values obtained from designs with values from benchmark networks, assuming that similarity to an optimal network implies optimality. To apply fractal analysis, three points of the retrofitted PFs for the considered case studies have been obtained:

(1) A min(*C*), min(*NRI*) point corresponding to the simultaneously minimum cost (*C)* and minimum reliability (*NRI*) WDS arrangement.
(2) A knee(*C*), knee(*NRI*) point corresponding to the knee cost and knee reliability *(NRI)* WDS arrangement.
(3) A max(*C*), max(*NRI*) point corresponding to the maximum cost (*C*) and maximum reliability (*NRI*) WDS arrangement.

For each case study, (1) and (3) have been chosen as the furthest points of the retrofitted PFs. Point (2) has been obtained as the intersection of the perpendicular line crossing the middle point of the segment connecting points (1) to (3) and the retrofitted PF of the case study. The location of point (2) has been corrected through a simple average between the coordinates of the intersection of the perpendicular line to the PF and the cost median of the case study PF. The latter guarantees a choice of point (2) in the knee of the considered retrofitted PF.

After selecting the three points of the retrofitted PFs for each case study, it was necessary to select an individual whose cost was equal or near to the selected PF point. For points (1) and (3) of the PF, the individuals with a cost equal to the one calculated with the methodology explained above were chosen. However, for the knee (2), there is no individual that had the same cost as the one obtained in the selection of the PF, so the individual that was selected was the one whose cost difference respect to the knee was the smallest. By obtaining the individuals of each case study, it was possible to perform the

fractal analysis since it has as input data information from the network, i.e., diameter, flow and HGL.

Through the application of the OPUS/NSGA-II methodology, 30 PFs of Hanoi and Fossolo were obtained. The repeatability of the algorithm in two networks of smaller magnitude allowed to verify the consistency of the methodology. Given the complexity of the algorithm and the greater network size, one PF was obtained in the cases of Balerma and Modena. Therefore, choosing the points of analysis from Hanoi and Fossolo implied obtaining the best approximation to the simple average of the coordinates of the minimum, knee and maximum points of the 30 PFs of these cases, for the best approximation was not necessarily in the same PF. The difference in the number of the PF allowed to best approximate the average of all the solutions of the OPUS/NSGA-II methodology in the cases of Hanoi and Fossolo. In contrast, the minimum, knee and maximum points were obtained from a single PF in Balerma and Modena.

### 3.5. Obtention of the Fractal Dimension

The purpose is to apply an existent methodology for calculating a network's fractal dimension; in particular, its box dimension. Subsequently, to determine whether there is correlation between the fractal dimension of a network and its optimality in terms of cost and resilience and to propose this measure as a metric to evaluate the desirability of a proposed network. This methodology will use results from the Pareto Front of optimal networks obtained with the OPUS/NSGA-II methodologies and seeks to assign a numerical value to networks on both ends of said front and at its *knee;* that is, lowest cost, highest resilience, and a mid-point, respectively. Ideally, this value could help designers know where their proposed designs are on the front, whether closer to some of the ends or to the inflexion point.

The topological coordinates of nodes along with some other parameter (piezometric head, total flow, and the sum of incoming and outgoing diameters at each node). The $z$-coordinate can be interpreted as a weight at node $j$ given by any of the following equations:

$$w_j = HGL_j \tag{5}$$

$$w_j = HGL_j \tag{6}$$

$$w_j = \left(\sum Q_{ij}\right) - D_j \tag{7}$$

where $HGL_j$ is the piezometric head value obtained by EPANET using instantaneous hydraulic simulation, $d_{ij}$ is the diameter of the pipe $i$ incoming or outgoing from node $j$, $Q_{ij}$ is the flow rate through the aforementioned pipe $i$ incoming or outgoing from node $j$, and $D_j$ is the demand at node $j$. The criteria will be refered to as the HGL, Diamater, and Flow Rate criterion, respectively.

The topology of the network along with one of these weights can be observed as points in three-dimensional space. To find the box dimension, the first step is to ensure that all the coordinates have the same order of magnitude. For example, if the $x$ and $y$ coordinates are thousands of meters and the piezometric heads are tens of meters, either the plane coordinates should be scaled down by a hundredth or the heads should be enlarged hundredfold. Next, define a boundary volume; that is, a cube in which envelopes all the points. This volume is then partitioned into equally sized boxes of side length \epsilon; this value will be changing from a larger one to a smaller one. For this experiment, the side length will be given by the total length of the cube divided by some increasing exponent of 2. In each iteration the objective function is the number of boxes needed to contain all the points.

The box dimension is formally defined as:

$$Dim(S) = \lim_{\varepsilon \to 0} \frac{\ln N(\varepsilon)}{\ln \varepsilon^{-1}} \tag{8}$$

where *Dim(S)* is the box dimension of some object *S*, $\varepsilon$ is the side length of the boxes used to cover the surface, and *N(ε)* is the number of such boxes needed to cover the whole surface [32]. Note that a box is only counted if some part of the object is contained within that box; in this case if some point is inside it. The limit evidently exists in this case since a finite number of points will need a finite number of boxes to be covered. Therefore, a power law equation can be used to simplify calculations:

$$D = \frac{\ln N(\varepsilon)}{\ln \ \varepsilon^{-1}} \tag{9}$$

where *D* is the fractal dimension. It is evident that equation 9 will be a linear plot on a logarithmic scale; this is the computational approach for calculating the limit in equation 8. With enough box sizes, the slope of the logarithmic plot will approximate the box dimension. To have a better estimation, the $R^2$ coefficient for the linear regression should be close to 1 [33].

A higher-level description of the algorithm is important to fully understand the process. The volume that encloses all the points should be a cube, to allow partitioning into cubical boxes. For every iteration, the cube of side length *L* is partitioned into boxes of side length $\varepsilon$. The algorithm then iterates over all the boxes and determines whether there are points in the boxes or not. Note the possibility of a point lying in the boundary between two or more boxes; to override this problem, the algorithm will erase all the points that have been counted to be in a box, therefore not counting several boxes for a single point. This also reduces the computational complexity, since fewer intersections of points and boxes must be verified. Finally, the number of boxes needed to cover all the points is saved and the algorithm iterates again with a smaller box size. This study uses at least ten values to plot $\ln N(\varepsilon)$ against $\ln \varepsilon^{-1}$ in order to have a trustworthy slope of that linear graph. The Python script stops iterating when an $R^2$ value greater than 0.99 is achieved given that a minimum number of iterations is made.

Having a finite and sometimes very low number of points is a limitation of the methodology. Therefore, an interpolation is carried out using a Python package (SciPy) in order to increase the number of points. Clearly, interpolating too many points off very few also generates error, so the general criterion suggested by the creators of the library is to interpolate at most $\frac{n^2}{4}$ points where *n* is the original amount of points. With this extended set of points, the analysis is carried out and the slope of the logarithmic curve is the fractal dimension. This interpolation also solves the issue of the linear slope being imprecise because the curve flattens. Bear in mind that the total amount of boxes needed to cover *n* points is at most *n*, so for small networks a few iterations would already cover every point in a single box. With the interpolation, the linear slope behaves well and reaches a desirable $R^2$ value before flattening.

The methodology will first be applied to four different networks (Hanoi, Fossolo, Modena, and Balerma) at three points: [min(*C*), min(*NRI*)], [knee(*C*), knee(*NRI*)], [max(*C*), max(*NRI*)]; that would mean obtaining nine different fractal dimensions for every network. This could show some pattern, especially if observed under the light of the *Classification* (explained below). In the search for a pattern within the many different networks on the Pareto Front for some networks, the three fractal dimensions of every single individual in that front were calculated: 500 individuals for Hanoi and 2000 individuals for Balerma.

### 3.6. Water Distribution System Classification

Understanding the intrinsic differences in the design alternatives for the min, max, and knee points of the PFs gives valuable information regarding the possible designs for a network. Additionally, studying the differences in the solutions for different types of networks allows one to analyse whether the proposed methodology works better for a particular type of WDS. Hence, the methodology proposed by Hwang and Lansey [34] has been used to characterize the PFs selected solutions for each network in this research.

This methodology allows one to classify WDSs according to their function and topology. Figure 1 shows the classification flowchart proposed by the authors for classifying the networks into six categories based on three parameters. More details on how the classification parameters are calculated and their thresholds can be found in Hwang and Lansey [34].

Regarding the functioning, WDS can be classified as transmission-dominated systems, which transport larger flows from the source to treatment plants and to distribution systems; or as distribution-dominated systems, which have smaller pipes and contain several service lines for single users [34]. As Figure 1 shows, this classification is performed by the length-weighted average pipe diameter ($\overline{D}$). Equation (10) shows how to calculate $\overline{D}$, where $D_k$ and $L_k$ are the diameter and length of pipe $k$, respectively, and $m$ is the number of pipes in the WDS.

$$\overline{D} = \frac{\sum_{k=1}^{m} D_k L_k}{\sum_{k=1}^{m} L_k} \tag{10}$$

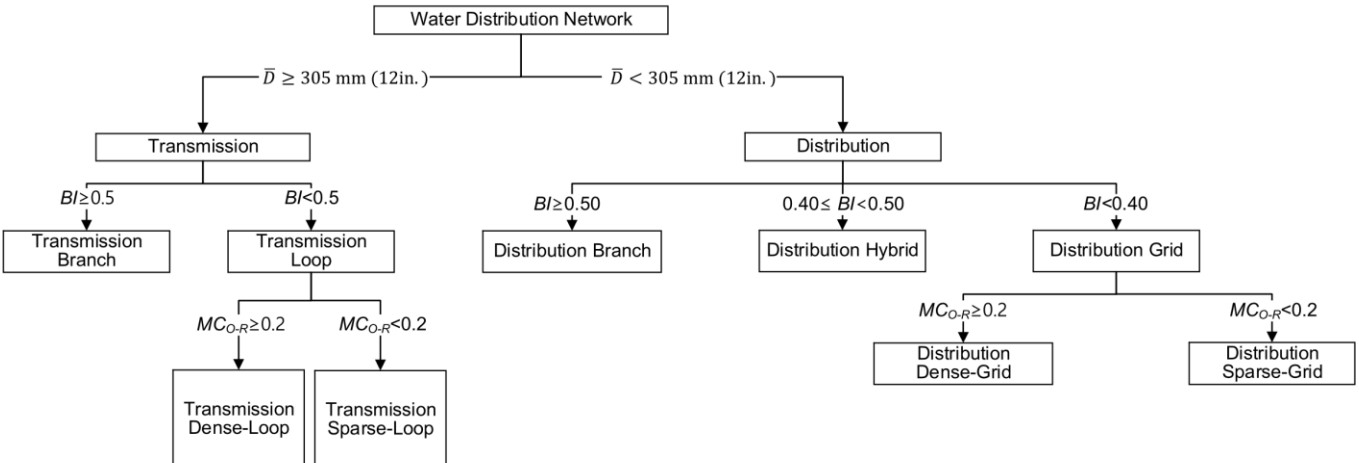

**Figure 1.** Water distribution network classification flowchart. $\overline{D}$ represents the length-weighted average pipe diameter, $BI$ is the Branch Index, and $MC_{O-R}$ is the Meshedness Coefficient for the reduced network. Source: Hwang and Lansey [34].

Concerning the topology, WDSs can be classified as loop (for transmission systems), grid (for distribution systems), or branch systems. Branch systems take water into smaller areas; grid systems are highly interconnected networks; loop systems are transmission mains consisting of multiple loops [34]. The Branch Index ($BI$) is used for differentiating branched or looped systems based on the ratio between branched edges to the addition of branched edges and the number of edges in a reduced network (skeletonized system) [34]. In this way, transmission systems can be classified as branched or looped, and distribution systems as branched, gridded, or hybrid (not branch-dominated nor grid-dominated). The $BI$ is calculated using Equation (11), where $e_b$ is the number of branched edges and $e_r$ is the number of edges in the reduced network [34].

$$BI = \frac{e_b}{e_r + e_b} \tag{11}$$

The transmission-looped and the distribution-gridded systems can also be cataloged as dense or sparse according to the Meshedness Coefficient ($MC$). Equation (12) shows how to calculate $MC$, which measures the network connectivity considering the number of loops in contrast to the maximum possible number [34].

$$MC = \frac{e - n + 1}{2n - 5} \tag{12}$$

where $e$ is the number of edges and $n$ is the number of nodes in the system. In this way, the parameters for the case studies' design alternatives have been calculated, and the systems have been classified to identify remarkable differences among the types of networks and the selected PFs solutions. Analyzing the outcomes from different types of networks enables understanding specific features from the OPUS/NSGA-II methodology and possible dissimilarities in the fractal analysis. Noteworthy, the layout of the systems does not change between designs in the same case study, just the diameters. Therefore, the topologic classification of each alternative is the same, but the functioning classification might differ according to the $\bar{D}$ parameter.

## 4. Study Cases

The methodology proposed in this paper has been tested in four well-known benchmarks networks: Hanoi, Fossolo, Balerma and Modena. Figure 2 shows all the chosen individuals for applying the fractal analysis algorithm. The chosen individuals have been obtained from the OPUS/NSGA-II retrofitted PFs.

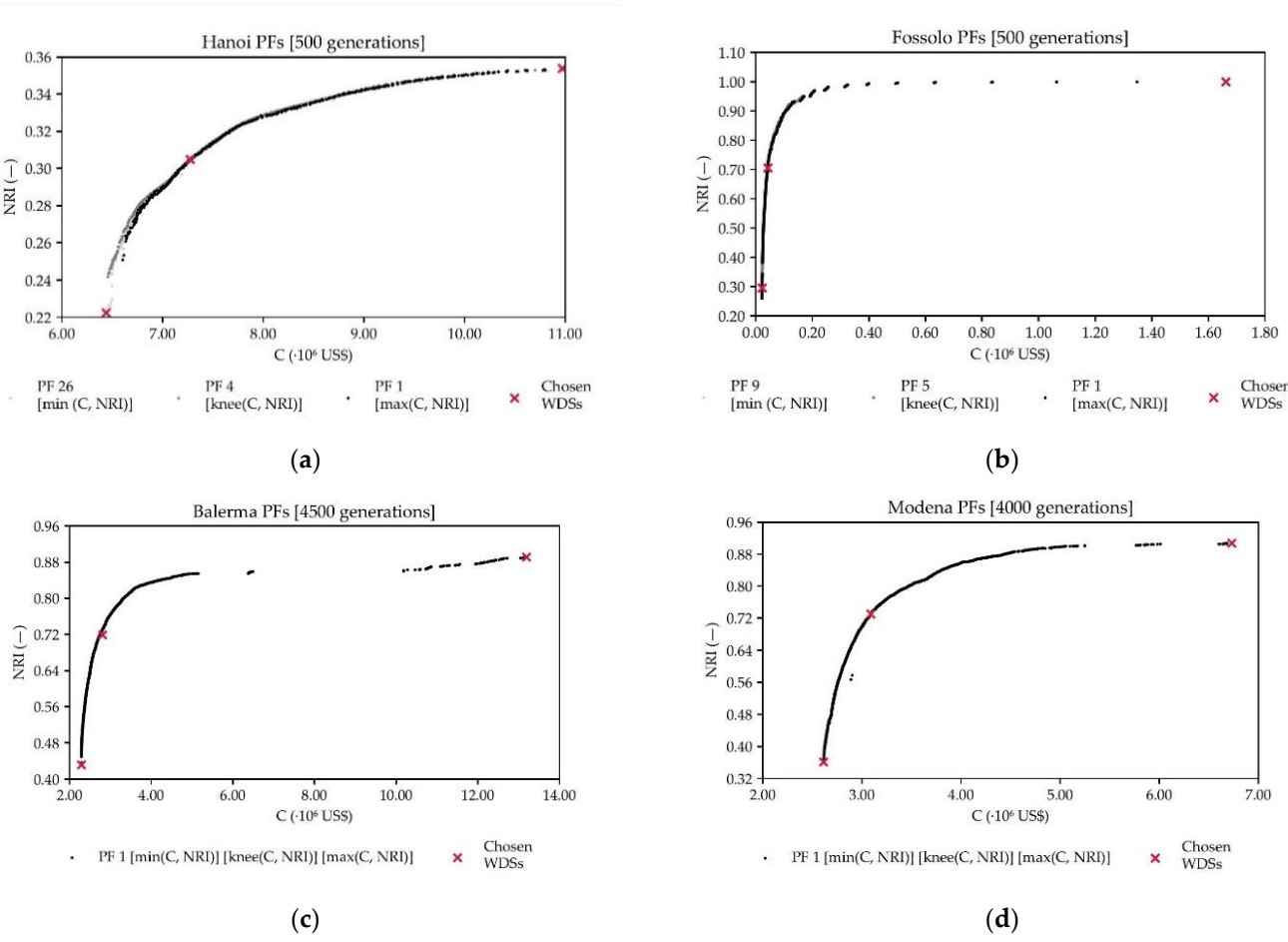

**Figure 2.** Selected individuals from retrofitted OPUS/NSGA-II PFs for the: (**a**) Hanoi WDS; (**b**) Fossolo WDS; (**c**) Balerma WDS; and (**d**) Modena WDS.

### 4.1. Hanoi

Hanoi was introduced by Fujiwara and Kang [35] and is a common water distribution network used to test optimization methodologies. The size of this network is medium, with 34 pipes, 32 nodes organized in three loops. There is no pumping station in the network, since there is one reservoir with an elevation of 100 m that supplies the entire system. The set of commercially available diameters is 12, 16, 20, 24, 30 and 40 inches. For the friction losses, the Hazen-Williams equation is used with a coefficient of C = 130 [35]. The

design has a minimum pressure restriction for all nodes of 30 m, but there is no velocity restriction. Figure 3 shows the HGL for the chosen WDS configurations for Hanoi (Table S1, Figure S1 in Supplementary Materials).

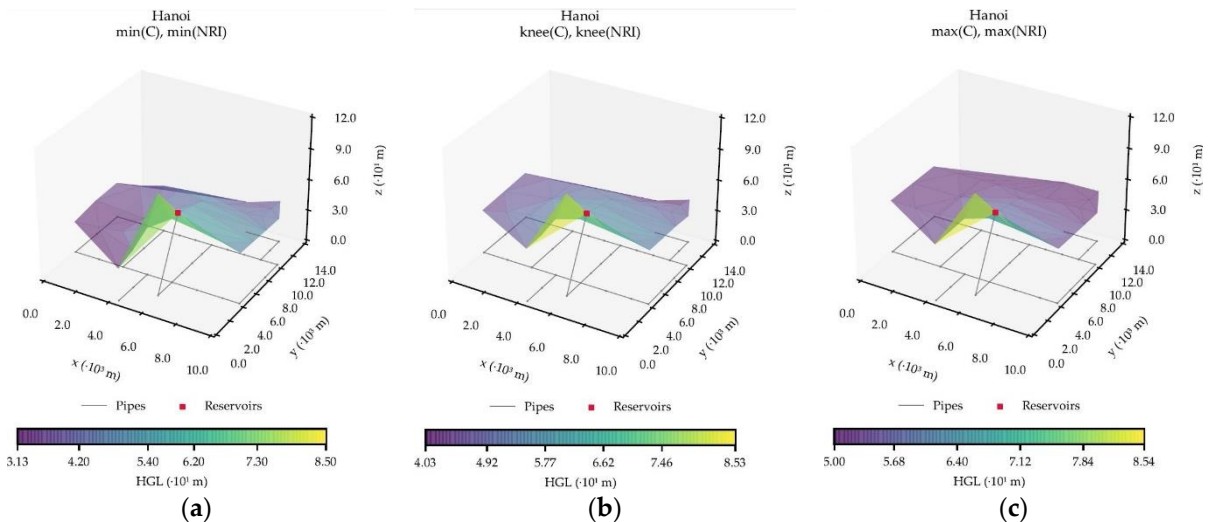

**Figure 3.** Hanoi WDS with the corresponding HGL surfaces for the: (**a**) [min(*C*), min(*NRI*)]; (**b**) [knee(*C*), knee(*NRI*)]; and (**c**) [max(*C*), max(*NRI*)] configurations.

### 4.2. Fossolo

Fossolo is a WDS based on the water distribution system for the city of Fossolo and was first introduced by Bragalli et al. [36]. The WDS consists of 58 pipes and 37 demand nodes supplied by one reservoir with an elevation of 121 m. Fossolo is an intermediate network with a search space of $7.25 \times 10^{77}$ possible design outcomes [18]. The set of commercially available diameters is 16, 20.4, 26, 32.6, 40.8, 51.4, 61.4, 73.6, 90, 102.2, 147.2, 184, 204.6 and 229.2 mm. The minimum pressure is $P_{min} = 40$ m and the maximum velocity in each pipe is 1 m/s. Figure 4 shows the HGL for the chosen WDS configurations for Fossolo (Table S4, Figure S4 in Supplementary Materials).

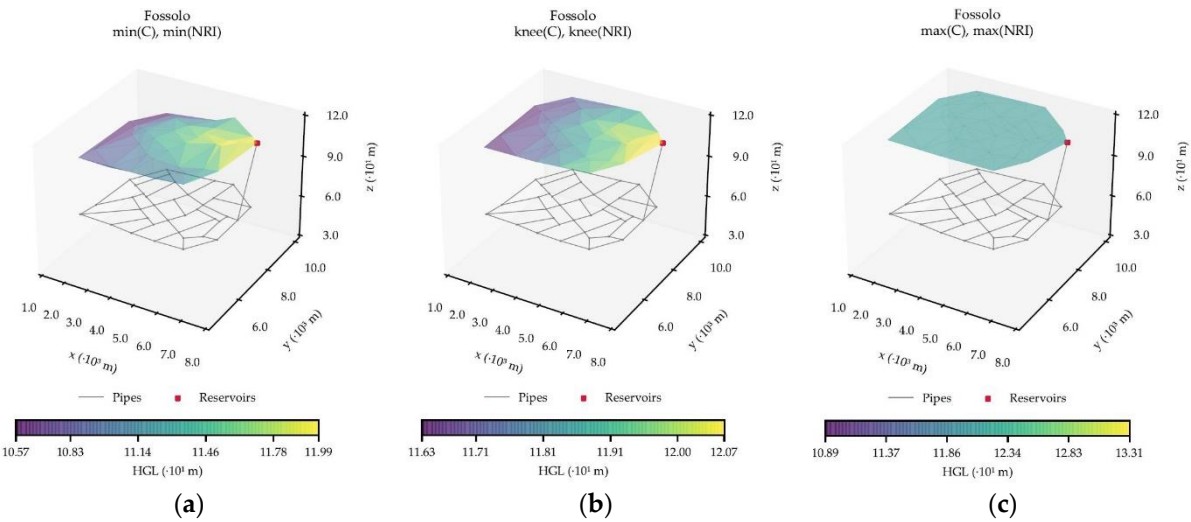

**Figure 4.** Fossolo WDSs with the corresponding HGL surfaces for the: (**a**) [min(*C*), min(*NRI*)]; (**b**) [knee(*C*), knee(*NRI*)]; and (**c**) [max(*C*), max(*NRI*)] configurations.

### 4.3. Balerma

Balerma is a WDS representing the irrigation system of the Poniente District located in Almeria, Spain [37]. The WDS consists of 454 pipes and 443 demand nodes supplied by four

reservoirs. Balerma has been used in the literature to study WDS optimization methodologies due to its large extension [37]. The network's commercially available diameters are 113, 126.6, 144.6, 162.8, 180.8, 226.2, 285, 361.8, 452.2, and 581.8 mm. Balerma has been calibrated through the Darcy-Weisbach head-loss expression, and all its pipes have been manufactured in PVC with an absolute roughness of $k_s$ = 0.0025 mm. The network includes a minimum pressure constraint of $P_{min}$ = 20 m [37]. Figure 5 shows the HGL for the chosen WDS configurations for Balerma (Table S2, Figure S2 in Supplementary Materials).

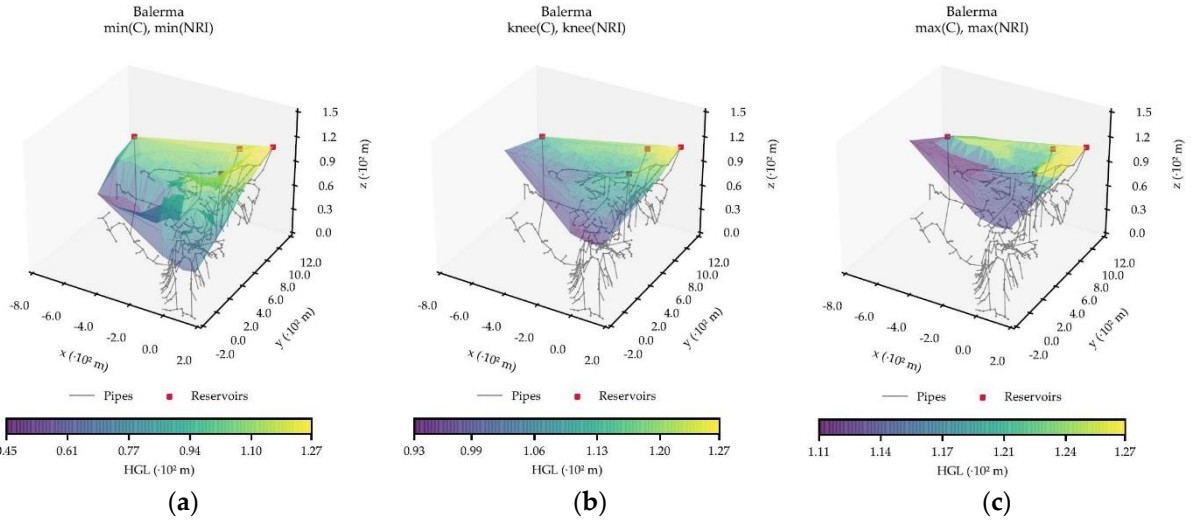

**Figure 5.** Balerma WDS with the corresponding HGL surfaces for the: (**a**) [min(C), min(NRI)]; (**b**) [knee(C), knee(NRI)]; and (**c**) [max(C), max(NRI)] configurations.

*4.4. Modena*

Modena is a WDS network presented by Bragalli et al. [36]. The WDS consists of 268 nodes, 317 pipes supplied by four reservoirs. Modena is a large network with a search space of $1.32 \times 10^{353}$ possible design outcomes [18]. For the friction losses, the Hazen-Williams equation is used with a coefficient of C = 130 for all pipes. The design has a minimum pressure restriction for all nodes of 20 m and a maximum velocity value of 2 m/s. Modena has 13 diameters available, whose values are between 100 mm and 200 mm. Figure 6 shows the HGL of the chosen WDS configurations for Modena (Table S3, Figure S3 in Supplementary Materials).

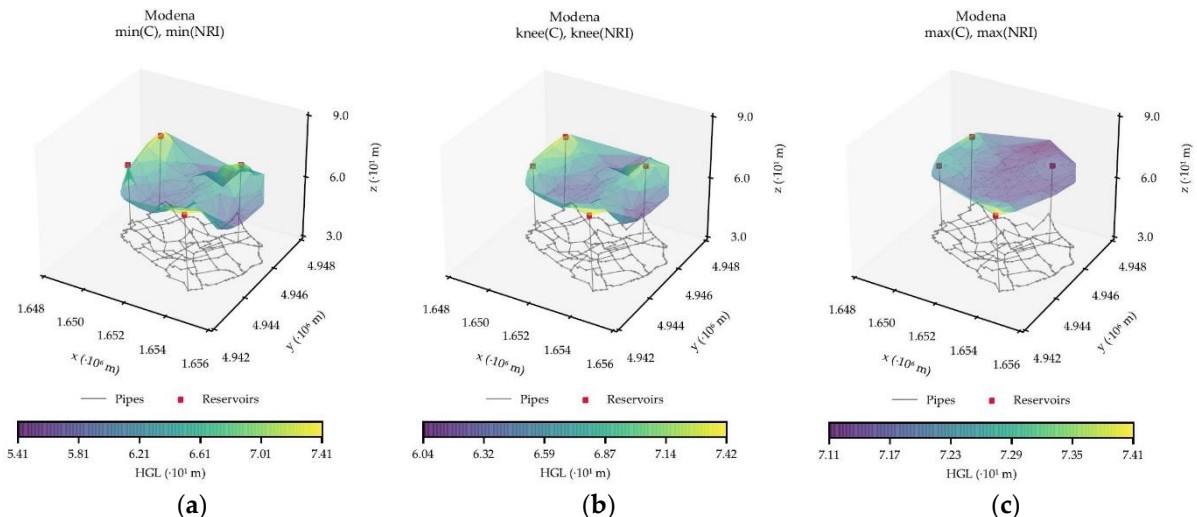

**Figure 6.** Modena WDS with the corresponding HGL surfaces for the: (**a**) [min(C), min(NRI)]; (**b**) [knee(C), knee(NRI)]; and (**c**) [max(C), max(NRI)] configurations.

Table 2 shows the characteristics of the selected networks, in terms of the number of reservoirs, pipes, pipe diameter options, pressure and velocity constraints, and the size of the search space. Table 3 shows the cost and NRI values of the selected networks.

**Table 2.** Characteristics of the study cases.

| Network | Reservoirs | Size | Pipes | Pipe Diameter Options | Search Space | Pressure Constraint | Velocity Constraint |
|---------|-----------|------|-------|----------------------|--------------|---------------------|---------------------|
| Hanoi | 1 | Medium | 34 | 6 | $2.8 \times 10^6$ | $P_{min} = 30$ m | No |
| Fossolo | 1 | Intermediate | 58 | 22 | $7.25 \times 10^{77}$ | $P_{min} = 40$ m | Yes |
| Balerma | 4 | Large | 454 | 10 | $1.00 \times 10^{455}$ | $P_{min} = 20$ m | No |
| Modena | 4 | Large | 317 | 13 | $1.32 \times 10^{353}$ | $P_{min} = 20$ m | Yes |

**Table 3.** Cost and NRI for the three selected points of each network: [min*(C)*, min*(NRI)*], [knee*(C)*, knee*(NRI)]*, and [max*(C)*, max*(NRI)]*.

| Network | Point | Cost ($) | NRI(-) |
|---------|-------|----------|--------|
| Hanoi | min(*C*), min(*NRI*) | 6,439,320.50 | 0.222 |
| | knee(*C*), knee(*NRI*) | 7,260,699.00 | 0.304 |
| | max(*C*), max(*NRI*) | 10,969,798.00 | 0.354 |
| Balerma | min(*C*), min(*NRI*) | 2,288,460.00 | 0.431 |
| | knee(*C*), knee(*NRI*) | 2,807,625.00 | 0.731 |
| | max(*C*), max(*NRI*) | 13,191,652.00 | 0.891 |
| Fossolo | min(*C*), min(*NRI*) | 23,046.98 | 0.295 |
| | knee(*C*), knee(*NRI*) | 43,330.23 | 0.715 |
| | max(*C*), max(*NRI*) | 1,661,922.50 | 0.999 |
| Modena | min(*C*), min(*NRI*) | 2,613,550.00 | 0.361 |
| | knee(*C*), knee(*NRI*) | 3,089,496.75 | 0.569 |
| | max(*C*), max(*NRI*) | 6,731,936.00 | 0.907 |

## 5. Results and Discussion

The type of water distribution network for each case study has been identified according to the classification flowchart (Figure 1) proposed by Hwang and Lansey [34].

Table 4 shows the parameters of the selected alternatives in the PFs for the studied WDSs and their classification.

**Table 4.** WDS classification, according to Hwang and Lansey [36]. $\overline{D}$ represents the length-weighted average pipe diameter, $BI$ is the Branch Index, and $MC$ is the Meshedness Coefficient for the reduced network.

| Network | Point | Class | $\overline{D}$ (mm) | $BI$ (−) | $MC$ (−) |
|---------|-------|-------|---------------------|----------|----------|
| Hanoi | min(*C*), min(*NRI*) | Transmission Dense-Loop (*TDL*) | 682.952 | | |
| | knee(*C*), knee(*NRI*) | Transmission Dense-Loop (*TDL*) | 750.144 | 0.438 | 0.333 |
| | max(*C*), max(*NRI*) | Transmission Dense-Loop (*TDL*) | 1016.000 | | |
| Balerma | min(*C*), min(*NRI*) | Distribution Branch (*DB*) | 173.259 | | |
| | knee(*C*), knee(*NRI*) | Distribution Branch (*DB*) | 189.439 | 0.770 | 0.052 |
| | max(*C*), max(*NRI*) | Transmission Branch (*TB*) | 446.902 | | |
| Fossolo | min(*C*), min(*NRI*) | Distribution Dense-Grid (*DDG*) | 37.542 | | |
| | knee(*C*), knee(*NRI*) | Distribution Dense-Grid (*DDG*) | 57.947 | 0.017 | 0.328 |
| | max(*C*), max(*NRI*) | Transmission Dense-Loop (*TDL*) | 409.200 | | |
| Modena | min(*C*), min(*NRI*) | Distribution Dense-Grid (*DDG*) | 145.779 | 0.033 | 0.331 |
| | knee(*C*), knee(*NRI*) | Distribution Dense-Grid (*DDG*) | 150.102 | | |

| Network | Point | Class | $\overline{D}$ (mm) | BI (−) | MC (−) |
|---|---|---|---|---|---|
| | max(*C*), max(*NRI*) | Distribution Dense-Grid (*DDG*) | 264.884 | | |

The Hanoi network uses large diameters for supplying the demand to users, even for the minimum cost design, i.e., minimum pipe diameters, so it is considered a transmission system. Regarding its functioning, it is classified as dense-loop, since it consists of three loops, as can be observed in the layout in Figure 3. Balerma network is an extensive irrigation system classified as branched due to its topology for supplying small areas. As previously mentioned, only the functioning classification might differ among PF solutions. Balerma and Fossolo networks have been the only ones for which a different class has been obtained. The alternatives corresponding to maximum cost and resilience have much larger diameters than those of the *min* and *knee* solutions, transforming the WDSs from distribution-dominated to transmission-dominated systems.

Regarding Fossolo and Modena networks, both WDSs were introduced by Bragalli et al. [36] and represent real distribution systems for the towns with the same name in Italy. As Figure 4 and Figure 6 show, these systems contain several service lines for supplying water to individual consumers. Hence, they are considered distribution-dominated systems, except for the *max* solution for Fossolo network because of larger pipe diameters. Regarding the topology, both WDSs have a small Branch Index (*BI*) and a large Meshedness Coefficient (*MC*), making them dense-gridded systems (dense-looped for Fossolo *max* alternative). This classification is due to the high interconnection of pipes within the water distribution networks for these Italian towns.

Table 5 shows the results of the fractal dimension for diameter ($\sum d_{ij}$), flow rate [$(\sum Q_{ij}) - D_j$] and pressure ($HGL_j$) criteria grouped by the resulting WDS classification of Table 4. In the case of $w_j = \sum d_{ij}$, the results tend to demonstrate an approach to a fractal dimension of $D = 2$, i.e., the expected value for a surface, according to fractal theory. Note that, for all the obtained classifications, $w_j = HGL_j$ produces fractal dimension values also approaching $D = 2$; consistent since an HGL energy distribution in a tridimensional plane corresponds to a surface. The behavior of $w_j = HGL_j$ resulting in outcomes closer to a surface is stronger in the case of Distribution Dense-Grids (*DDGs*), as it could happen that these types of networks closer resemble a surface than Transmission Dense-Loops (*TDLs*) or Distribution and Transmission Branches (*DBs and TBs*). The max(*C*), max(*NRI*) value of $D = 0.874$ for Fossolo may represent outliers given by a solution in the retrofitted OPUS/NSGA-II PF, which is deviated from the cost (*C*) and network resilience index (*NRI*) of most individuals, as seen in Figure 4c. Finally, in the case of $w_j = (\sum Q_{ij}) - D_j$, values are equal for each revision individual in the case of *TDLs*, *DBs*, and *TBs*, but show slight variations in the case of *DDGs*. An equal $D$ in the case of $w_j = (\sum Q_{ij}) - D_j$ for TDLs, *DBs* and *TBs* may be a result of an attained limit for $D$ through this revision criteria in optimized networks classified into these categories.

**Table 5.** Fractal dimension *(D)* for the revised criteria $w_j$.

| Classification | Network | $w_j$ | min(*C*), min(*NRI*) | knee(*C*), knee(*NRI*) | max(*C*), max(*NRI*) |
|---|---|---|---|---|---|
| Transmission Dense-Loop (TDL) | Hanoi | $\sum d_{ij}$ | 1.929 | 1.934 | 1.891 |
| | | $HGL_j$ | 1.829 | 1.826 | 1.946 |
| | | $(\sum Q_{ij}) - D_j$ | 1.020 | 1.020 | 1.020 |
| Distribution Branch (DB), Transmission Branch (TB) | Balerma | $\sum d_{ij}$ | 1.857 | 1.807 | 1.798 |
| | | $HGL_j$ | 1.798 | 1.798 | 1.791 |
| | | $(\sum Q_{ij}) - D_j$ | 1.798 | 1.798 | 1.798 |
| Distribution Dense-Grid (DDG) | Fossolo | $\sum d_{ij}$ | 1.950 | 1.829 | 1.829 |
| | | $HGL_j$ | 2.033 | 2.047 | 0.874 |
| | | $(\sum Q_{ij}) - D_j$ | 1.831 | 1.835 | 1.839 |

| | | | | |
|---|---|---|---|---|
| Modena | $\sum d_{ij}$ | 1.936 | 1.941 | 1.875 |
| | $HGL_j$ | 2.040 | 2.044 | 2.082 |
| | $\left(\sum Q_{ij}\right) - D_j$ | 1.976 | 1.966 | 1.940 |

Results in Table 5 serve as an inspection approach for guiding the fractal analysis of the selected WDSs. The objective of the next step has been to investigate if tendencies in a WDS can be generalized for a complete set of solutions, i.e., a retrofitted OPUS/NSGA-II Pareto Front. Fractal analysis metrics have been applied to the set of individuals of a retrofitted OPUS/NSGA-II PF in the cases of Hanoi and Balerma. Figure 7 shows the results of applying the presented fractal analysis criteria of WDSs on a Hanoi retrofitted OPUS/NSGA-II PF. Figure 8 shows the results of applying the presented fractal analysis criteria of WDSs on a Balerma retrofitted OPUS/NSGA-II PF.

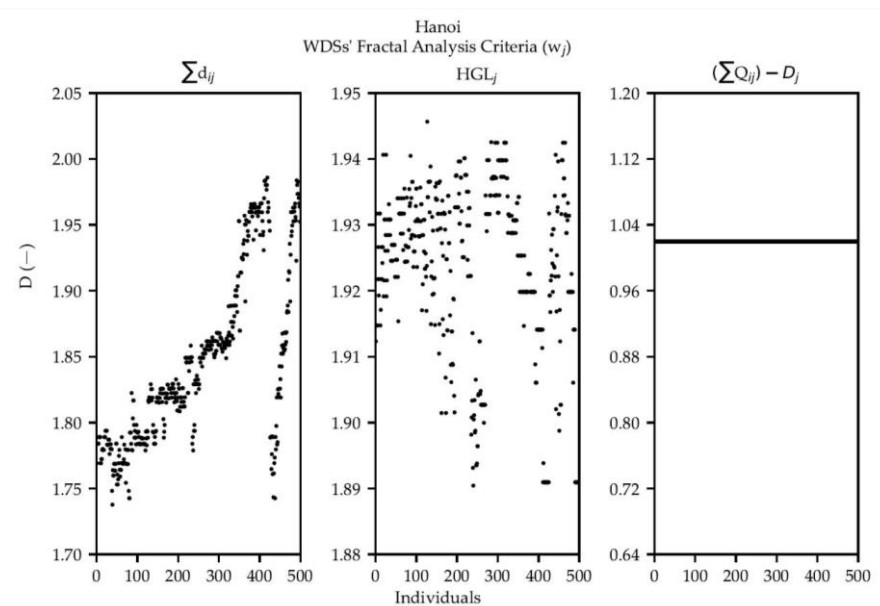

**Figure 7.** WDSs' fractal analysis criteria ($w_j$) applied to a retrofitted OPUS/NSGA-II Hanoi PF.

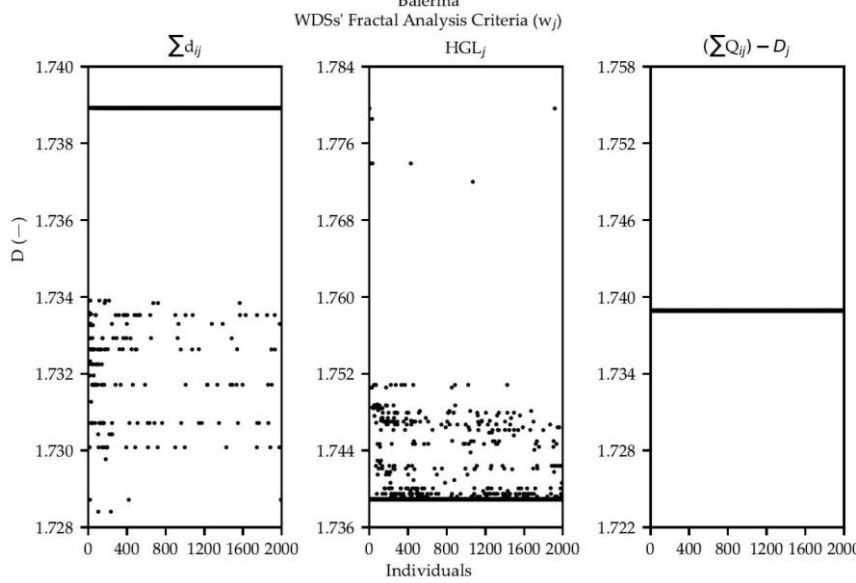

**Figure 8.** WDSs' fractal analysis criteria ($w_j$) applied to a retrofitted OPUS/NSGA-II Balerma PF.

The definition of the box dimension suggests that a value of D close to 2 means that the object resembles a surface. Observing results over the three points of interest in the four case studies, all but four of the values are around 2. This line of thought implies that in almost every case the points are not greatly scattered; but rather homogenously laying in space around some plane. This is consistent with the methodology, bearing in mind that a surface interpolation is carried out. The analyses of the PFs further solidify this idea and demonstrate another interesting tendency: some criteria have a dimension to which most of the networks tend. Figure 7 shows it for the Flow Rate case of Hanoi and the three criteria for Balerma also expose this in Figure 8. Although by no means conclusive, most optimal networks are similar to each other under the light of these metrics. This is not as visible for Hanoi as it is for Balerma; but bearing in mind the size of one and the other, the idea that there is a value that describes the vast majority of networks should not be abandoned.

Out of the three criteria, the HGL one is of greater interest as the dimension can directly correlate to optimality. Ideally, the piezometric head values at the nodes should vary very little since low energy losses are desired. If this is the case, the points will lay very close to a single plane and the calculated box dimension should be around 2. If the contrary happened, points would be scattered in space, sometimes very distant from each other and the dimension would be lower, tending to 1 or even 0. Table 5 shows that Modena has a very homogeneous HGL in all cases and so does Fossolo, except for one. When looking into the head values for this network, it turns out that all of them are the same. It is perhaps the case that the network in fact resembles a line (1 dimensional) when laying perfectly on the same plane; this point can be observed as an outlier, since it is almost never the case that all head sizes are the same.

The Flow Rate criterion exposes a pattern but not all that interesting. All the networks in Hanoi and Balerma have exactly the same fractal dimension (1.02 and 1.738, respectively) according to Figures 7 and 8. The pattern is even stronger: the three points of interest in Balerma from Table 5 obtained the same value among each other with a finer interpolation. Values for Modena and Fossolo also have a very small variation in the order of the thousandths. It seems that this criterion is not very telling of the differences between optimized networks and is perhaps not worth looking at more in depth.

The Diameter criterion, as the HGL did, tends to a value of 2 in Table 5. This is again consistent with the methodology, since a surface of points is interpolated. This could perhaps suggest that homogeneity in the distribution of the values is desirable. When looking into the PFs, the results are somewhat different. In the case of Hanoi, for the Diameter criterion, Figure 7 shows increasing dimension values for the first individuals. That is, the fractal dimension ($D$) tends to increase when a higher $C$, higher $NRI$ solution is revised. Towards the end, around individual 420, $D$ values get abruptly smaller, to finally increase towards the end of the retrofitted PF. The latter behavior can be explained by the fact that lower to middle cost solutions tend to employ mixed diameter values for a given WDS configuration, i.e., the solutions demonstrate more dispersion in space with $\sum d_{ij}$ as the z-coordinate. As the diameters become more homogeneous, the dimension increases but at some point, the configuration becomes again of mixed diameters although greater in size. At this point it is worth noting that the dimension does not depend on the location of the set of points in space, but their positions relative to each other. That is, increasing the diameter at every node does not affect the dimension; but increasing some and leaving others the same does. This tendency can be a particular case of a *TDL*, as is Hanoi. It can be further explored to understand differentiating patterns between optimized designs in these types of networks.

In the case of the $w_j = HGL_j$ criterion for Hanoi, Figure 7 shows that the fractal analysis methodology is consistent, as solutions for the fractal dimension approach to a value of $D = 2$. Finally, note that the weight $w_j = \left(\sum Q_{ij}\right) - D_j$ in Figure 7 produces a single value of *D = 1.02* for all considered individuals, condition that may be an indicator of equal flow rate distribution in all selected networks in a retrofitted OPUS/NSGA-II PF. A same

outcome for all individuals may indicate that all networks provide an equal flow rate dispersion, that is, they demonstrate an equal performance in terms of demand supplied to users. All the fractal analysis criteria applied to Hanoi as notions for the design of TDLs are preliminary but represent a further step in optimized network design. It is necessary to examine an elevated number of optimal network designs in the same category to find conclusive values of *D*, which can help to design optimal water networks of this type. It is important to obtain results by WDS classification, as different topological arrangements of networks can demonstrate different behaviors in fractal dimension (*D*), as Figures 7 and 8 exhibit.

It is also important to note that Hanoi is a very small and symmetric network, which is not as desirable in fractal analysis since fractals are by nature chaotic shapes. It is picked because analyzing it has a very low computational cost. On the other end is Balerma, a network more than tenfold larger in number of nodes. The analysis of this Pareto Front shows clearer patterns that have already been mentioned. The most interesting one is the fact that most individuals tend to some fractal dimension for each of the criteria. Not only that, but the value is in fact the same for the three criteria (1.739). That is very telling: most optimal designs for this network have a very similar distribution in space regardless of the criterion used. This is not tremendously surprising, since two of the three conditions are in fact the same (the topological coordinates). However, it does imply that the three criteria are related in its distribution in space; there is some proportionality between the diameter ($\sum d_{ij}$), HGL ($HGL_j$) and flow rate [$w_j = (\sum Q_{ij}) - D_j$] criteria, at least for the Balerma network. It is therefore something worth looking at using other networks of a similar size. However, the question is why some of the networks do not converge to that same value. When observing Figure 8, it is evident that individuals along the whole front are out of that limit value; from the end with lower cost and lower *NRI* to the other end with the higher values. Therefore, this condition might not necessarily be related to the position on the front but to some other characteristic of the network.

For the diameter criterion ($\sum d_{ij}$), there is a gap between the limit value and all the points out of it, which are all under 1.739. In the case of the HGL criterion it is different, all points are above the limit value, most of them close and only a couple farther away. Bearing in mind the fact that a greater value is desirable for this criterion, another question is whether the outlier networks are the best alternative to minimize energy losses. A closer look at the data revealed that most designs that did not have the limit dimension for one criterion did have it for the other, which suggests very specific characteristics of the designs are to blame for these outliers. Once again it is important to mention that these differences are of the order of hundredths.

## 6. Conclusions and Future Work

One of the most used multi-objective algorithms for the WDS optimization problem is the NSGA-II; however, this algorithm implements a high computational cost to reach an acceptable solution. For this reason, different methodologies have been sought to obtain a near-optimal solution in a shorter computational time. One of the methodologies that meet this objective are those based on energy criteria. OPUS is a design methodology based on the HGL criterion that obtains a near-optimal design of a WDS in a shorter computational time. Nevertheless, WDS design cannot only be based on cost reduction, but resilience and reliability must also be considered. For this reason, the OPUS/NSGA-II algorithm was developed with the objective of minimizing the capital cost of the system and maximizing its reliability.

To understand the characteristics of optimal WDS and investigate criteria that could be used for reducing the computational time of the algorithms, near-optimal solutions from the PFs obtained with the OPUS/NSGA-II methodology have been studied for four benchmark networks. The optimal energy dissipation from these systems, which resembles a surface, has been analyzed using the box-counting dimension, a usual method for studying fractals. Likewise, the distribution of flows and diameters in the optimized

systems have been studied through this fractal measure. First, the water distribution networks have been classified according to their topology and functioning using a widely employed methodology to comprehend the differences in the methods employed among the case studies. In this way, specific features have been identified for transmission-dominated (dense-looped and branched) and distribution-dominated systems (dense-gridded and branched).

The study of the box dimension has been demonstrated to be greatly sensitive to completely understand and validate the methodology. It has been demonstrated that the methodology is consistent with the mathematical theory along all the networks. Furthermore, the algorithm has proven to be time efficient, as it was run over large networks in a matter of minutes due to lesser computational complexity: for example, a run over Balerma took around 5 min for a computer with 12 GB of RAM and a Core i7 8th gen. processor. When looking back to the algorithm, its time complexity is $O(n^3)$, which is polynomial and desirable. The results do show tendencies, but these are of a very small order; in general, it is safe to suggest that the fractal dimensions of optimal designs for a given network are almost the same. Although this study only analyzed optimal networks, Jaramillo [31] also found fractal dimensions, although with a slightly different methodology, of non-optimal networks and did not observe the patterns here exemplified. That suggests that the limiting value could be unique to the optimal networks. Therefore, this paper has successfully developed a consistent methodology with a low computational cost that can be integrated to more costly algorithms as an optimality criterion that can reduce the overall running time.

This procedure is promising in the search for design criteria with a very low computational cost. It could be applied in the optimization process of networks: once a handful of optimal networks are known and given that their fractal dimensions are almost the same for the different criteria, evaluating whether a design option is optimal could be conducted easily by calculating its dimensions. In the search for the optimum, this could serve as a type of gradient method where changes are conducted in order to approximate the dimension to said value. Said integration to other methods is one of the challenges for the future.

Even though the methodology has demonstrated interesting and consistent results, its application in this study was limited. Analyzing more PFs and networks pertaining to different types is important in the search for more consistent patterns. For instance, it would be important to confirm whether Hanoi's results are due to it being a Transmission Dense-Loop or if it is more related to its size or topology. It would also be key to confirm whether the behavior demonstrated by Balerma is unique to its class of networks or if it has more to do with the size. This study opens the door for research on the topic by providing a trustable methodology at a low computational cost [38,39]

Future research could adapt this proposed methodology to analyze networks with pumping stations. In addition, this methodology could be used for the optimization of pumping patterns, as well as for sectorization and location of storage tanks, among others. This was not conducted in this research since it was not the main objective; however, with some modifications of the code it would be possible to cover different problems that are present in the optimized design of water distribution systems.

**Supplementary Materials:** The following supporting information can be downloaded at: https://www.mdpi.com/article/10.3390/w14233795/s1. Table S1: Hanoi, [min(C), min(NRI)], [knee(C), knee(C)], [max(C), max(NRI)] individual choices for analysis of optimal WDS configurations obtained through the OPUS/NSGA-II Methodology; Table S2: Balerma, [min(C), min(NRI)], [knee(C), knee(C)], [max(C), max(NRI)] individual for analysis of optimal WDS configurations obtained through the OPUS/NSGA-II Methodology; Table S3: Modena, [min(C), min(NRI)], [knee(C), knee(C)], [max(C), max(NRI)] individual for analysis of optimal WDS configurations obtained through the OPUS/NSGA-II Methodology; Table S4: Fossolo, [min(C), min(NRI)], [knee(C), knee(C)], [max(C), max(NRI)] individual for analysis of optimal WDS configurations obtained through the OPUS/NSGA-II Methodology; Figure S1: Hanoi pipe diameter choices in the final WDS

configuration obtained through OPUS/NSGA II for the (a) [min(C), min(NRI)] (b) [knee(C), knee(C)] and (c) [max(C), max(NRI)]; Figure S2: Balerma pipe diameter choices in the final WDS configuration obtained through OPUS/NSGA II for the (a) [min(C), min(NRI)] (b) [knee(C), knee(C)] and (c) [max(C), max(NRI)]; Figure S3: Fosolo pipe diameter choices in the final WDS configuration obtained through OPUS/NSGA II for the (a) [min(C), min(NRI)] (b) [knee(C), knee(C)] and (c) [max(C), max(NRI)]; Figure S4: Modena pipe diameter choices in the final WDS configuration obtained through OPUS/NSGA II for the (a) [min(C), min(NRI)] (b) [knee(C), knee(C)] and (c) [max(C), max(NRI)], Table S5. Values of each parameter for the WDS classification. The fractal analysis code can be observed in: https://github.com/FeWiesner/Fractal_Wiesner (accessed on1 October 2022).

**Author Contributions:** Conceptualization, J.S. and C.S.; methodology, J.S., C.S., M.A.G., C.O., S.G. and F.W.; software, S.G. and F.W.; validation, J.S., C.S., M.A.G. and C.O.; formal analysis, M.A.G., C.O., F.W. and S.G.; investigation, M.A.G., C.O., S.G. and F.W.; writing—original draft preparation, M.A.G., C.O., S.G. and F.W.; writing—review and editing, J.S. and C.O.; supervision J.S., C.O. and M.A.G. All authors have read and agreed to the published version of the manuscript.

**Funding:** This research received no external funding.

**Data Availability Statement:** The data presented in this study are available on request from the corresponding author.

**Conflicts of Interest:** The authors declare no conflict of interest.

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
