# Peer review of "On the Evolution of the Optimal Design of WDS: Shifting towards the Use of a Fractal Criterion"

_water, doi:10.3390/w14233795_

Round 1

Reviewer 1 Report

-        Ref # 10 is not the right format, the authors list is not complete

-        It is understandable that the area is probably your landmark, with not a lot of work on it from other researchers, but it seems there are a lot of self-citations to the authors on the reference list. I would suggest diversifying a little bit to add other authors in the field as well.

-        This is a cool approach, but I would curtail paragraphs like those from line 57 to 76 and put more effort in explaining the paragraph right after.

-        In the Abstract, please explain what improvements you achieved, such as how many % lower cost of design or analysis? How much faster in obtaining the results compared to NSGA-II or how many fewer iterations to achieve the optimal solutions?

-        I am also not sure if section 2 (background) should be a separate section, and why not a part of the introduction.

-        It seems like the approach is useful for design as it uses the hydraulic head in different locations of the system. My question is if the approach is applicable for pump scheduling optimization for example or valve setting optimization of finding the right location for an elevated storage tank? And if so, please explain how.

-        Is this approach an “add-on’ to NSGA-II? Please clarify

-        Have you ever applied this method to a real- world distribution system? If not, please explain why and what the obstacles are?

-        I believe that the headings should be capitalized for word.

-        The conclusion is very lengthy and more than half of it spends on literature review and the importance and abstract. Please report on what was exactly achieved and what will be later on if you have future plans with this method.

Reviewer 2 Report

The work is extremely complex. Many cooperating algorithms were used, and finally the fractal analysis was introduced, which is to evaluate the current solution of the water distribution system using three parameters. The results of optimization on exemplary water supply systems are presented.

After reading the article, the following comments and questions arise:

In the opinion of the reviewer, the article should indicate for which water intake conditions optimization calculations for the water distribution system are carried out. Variable conditions of water intake mean that pressure losses in individual water pipes are very different during the day, which can affect the selection of the diameters. Not all diameters of pipes are selected for the conditions of maximum water consumption, e.g., on transit pipes to tanks.

The important question is whether the water distribution system is to provide water for firefighting purposes. In this case, the diameters of the water pipes are often determined by the flow and pressure requirements for fire conditions, greatly complicating the design of water networks.

The authors write about the algorithms that were used, but the computer program in which these algorithms were implemented was not mentioned. The considerable complexity of the optimization procedure justifies the discussion of this issue in this article. The methodology of cooperation between the proprietary application and the EPANET program was also not mentioned.

There are a significant number of self-citations in the article. In the opinion of the reviewer, bibliographic items no. 10 and 25 are not justified.

References 8 and 29 refer to different editions of the EPANET program documentation. Is it justified by the differences in these documents?

Chapter 3. Study cases should be number 4.

Reviewer 3 Report

The paper studies different methodologies for the optimal design of water distribution systems, involving hydraulic concepts that should permit the control of the optimization design process. The paper is well written and i think it should be published in Water after taking into account the following comments:

- Lines 481, 489 and 594: please fix the sentences showing compilation errors.

- Lines 100-107: Please refer to the Sections as organization of the manuscript. For example, "In Sect. 2 an explanation of the use of each method is given.... Sect. 5 closes the paper with Conclusions and future work ".

Reviewer 4 Report

This paper aims to understand the characteristics of near-optimal solutions using three designs from the OPUS/NSGA-II Pareto fronts of four distinct networks. This paper finally proposes a design criterion based on the hydraulic gradient line box dimension.

The logic of the manuscript is clear, the arguments are clear, and the citations are appropriate. It has explained the meaning of OPUS, NSGA-II, OPUS/NSGA-II, and fractal analysis in detail. However, the following details need to be modified or given a reasonable explanation.

1. The Optimal Design of WDS proposed in this paper has been verified by four well-known benchmark networks, but all of them are no pumping stations. The design criterion proposed in this paper should be limited, or the application of this standard to the pumping station should be explained.

2. There are too many errors in the heading. The confusion starts at line 344, and then lines 395, 474, 525, and so on are all wrong.

3. In line 532, Figures c and d are inconsistent with figures a and b, and do not indicate the PF values of the knee (C, NRI) and max (C, NRI).

4. In lines 546, 571, 557, and 583, since the same pipe network is verified, why are the upper and lower limits of HGL scales inconsistent? Will different scales be used to misunderstand HGL for readers?

5. For the Hanoi and Balerma models, why there is no limit on speed in pipe network design, please provide reliable theoretical support (reference) or a reasonable explanation.

6. In Table 4, no explanation of BI and MC is found in the paper.

7. In lines 481, 489, and 594, references are missing.

8. The article gives an abbreviated explanation for Transmission Dense-Loop, but almost every time TDL appears, it is explained. I don't understand the meaning of the abbreviation.

9. In lines 778-781, the algorithm has proven to be time efficient. As the key factor of algorithm design, the solving speed should not be ignored. The manuscript should be supported by specific comparative data, not simply saying that the solution time is 5 minutes.
